Road ecology of a Chihuahuan Desert snake community: size-based mortality sets the stage for evolutionary change in a widespread pitviper

http://orcid.org/0000-0001-5883-5324 Hughes Daniel F. 1 2 dfhchameleon@gmail.com
1 Department of Biology, Coe College , Cedar Rapids, Iowa , United States
2 Department of Biological Sciences, Tarleton State University , Stephenville, Texas , United States
Brygadyrenko Viktor
Electronic publication date: 2025 Aug 12
Publication date: 2025
Volume: 13
Electronic Location ID: e19871
Received 2025 Apr 8; Accepted 2025 Jul 17
Copyright: © 2025 Hughes
Copyright year: 2025
Copyright holder: Hughes
License: This is an open access article distributed under the terms of the Creative Commons Attribution License, which permits unrestricted use, distribution, reproduction and adaptation in any medium and for any purpose provided that it is properly attributed. For attribution, the original author(s), title, publication source (PeerJ) and either DOI or URL of the article must be cited.
License URL: https://creativecommons.org/licenses/by/4.0/

Keywords: Body size, Crotalus atrox, Dead-on-road, Directional selection, Non-random mortality

Funding: The author received no funding for this work.

==============================
Snakes are significant predators in many ecosystems, but high rates of road mortality threaten to diminish their ecological contributions. Documenting species-specific and demographic patterns of road use is crucial for understanding potential impacts, information that can be leveraged for implementing mitigation measures. I investigated the road ecology of a snake community in the Chihuahuan Desert of southern New Mexico, USA. Nocturnal road-cruising surveys were performed three times per month from April to November 2017 along a 37-km stretch of road, which produced 101 snake encounters of 10 species. On average, 4.2 snakes were encountered per survey with no individuals recaptured, equating to a density of 0.057 snakes per kilometer. Seasonal activity patterns indicated a peak in snake encounters from August through October. Standardized data were combined with opportunistic surveys conducted along the same stretch of road from 2014 to 2016. Overall, road mortality was considerable, with 34% of all snakes found dead-on-road (DOR). Across all species, males and females exhibited similar proportions of DOR while juveniles had a lower DOR rate. The Western Diamondback Rattlesnake (Crotalus atrox) was the most frequently observed species across all surveys (48% of all encounters) and 33% of these observations were found DOR. Models for adult C. atrox revealed that longer body sizes were linked to road mortality, where every 1 cm increase in total length increased the probability of a vehicle collision by almost 10%. Preliminary simulations indicated that directional selection against larger body sizes could reduce the mean size in this population by nearly 12 cm in 50 generations (or about 165 years in this species). Road mortality may have set the stage for evolutionary change in a widespread and ecologically important pitviper. Because roads and vehicle traffic will continue to expand globally, efforts to mitigate road-based effects on snakes are essential to implement before major ecological and evolutionary impacts are felt.

Introduction

Snakes are critical components of many ecosystems, serving as predators that can help regulate prey populations (Shine & Madsen, 1997; Nowak, Theimer & Schuett, 2008). In this capacity, snakes contribute directly and indirectly to the control of biodiversity in ecological communities (Emery et al., 2021). For example, a shift in snake predation on an island led to the collapse of an entire reptile community, largely through direct effects (Lin et al., 2023). Snake predation can also produce changes within prey populations through selection. For example, lizard populations exposed to snake predation exhibited warmer body temperatures, longer limb lengths, and faster sprint speeds compared to control populations (Yuan et al., 2021). Snakes not only influence prey populations in a multitude of ways, but they have also become important bioindicators of several environmental issues (Beaupre & Douglas, 2009), including pollution (Hoang et al., 2021), contamination (Haskins et al., 2021), and urbanization (Lettoof et al., 2021). Understanding the factors that influence snake populations is thus consequential for a range of ecological issues and is particularly urgent in the face of growing challenges that pose serious threats to snakes (Todd, Willson & Gibbons, 2010). Among the many challenges snakes face, human-driven changes are perhaps the most significant, particularly the construction and use of roads (Rosen & Lowe, 1994).

Roads have an outsized impact on snakes as they present a dual-edged threat (Andrews & Gibbons, 2005): they act as barriers to movement that limits population connectivity (Clark et al., 2010; Tipton, Vázquez-Diosdado & DeSantis, 2023) and also as sites of high mortality due to vehicles striking snakes (McCardle, Fontenot & Lutterschmidt, 2022; Meshaka, Mazzotti & Rochford, 2024; Szabolcs et al., 2024). These road-based effects can reduce snake populations and thereby disrupt the ecological services they provide (Arzamendia et al., 2024). For example, one study showed that even low rates of road mortality could increase the extinction probability in a population of snakes from 7.3% to 99% in 500 years (Row, Blouin-Demers & Weatherhead, 2007). Another study found that the presence of a major road led to a significant reduction in gene flow within a snake population (Herrmann et al., 2017). Moreover, road mortality affects some species and demographic groups disproportionately (Bonnet, Naulleau & Shine, 1999), which can exacerbate declines in already dwindling populations (Hartmann, Hartmann & Martins, 2011; Wagner, Brune & Popescu, 2021; Morelli et al., 2024). Identifying the species most at risk, their road-related behaviors, and any demographic consequences associated with road mortality is essential for designing effective mitigation strategies (Medrano-Vizcaíno & Espinosa, 2021; Jones et al., 2024). Without targeted conservation efforts, the expansion of global road networks will likely exacerbate the challenges faced by snakes to maintain viable populations (Trombulak & Frissell, 2000; Grilo et al., 2021).

Most snakes are cryptically patterned, many species are considered rare, and almost all exhibit secretive behaviors, factors that render them difficult to detect, let alone study in detail (Steen, 2010; Durso, Willson & Winne, 2011). Road-cruising surveys therefore present a powerful tool for studying snakes (Sullivan, 2012), providing unprecedented opportunities to document activity patterns, species composition, and mortality rates in a relatively understudied group (Klauber, 1939; Fitch, 1949; Dodd, Enge & Stuart, 1989; Rosen & Lowe, 1994; Langen et al., 2007). By covering large areas in a standardized manner, road-cruising surveys generate data on both live and dead-on-road (DOR) individuals (Köhler, Ceeño-Vázuez & Beutespacher-García, 2016), offering valuable insights into population structure, movement patterns, and the impacts of roads on local snake communities (McDonald, 2012; Jochimsen, Peterson & Harmon, 2014). Such data are critical for understanding species-specific vulnerabilities and guiding conservation efforts aimed at mitigating the effects of roads on snake populations (Boyle et al., 2021; Macpherson et al., 2021).

In this study, I set out to document patterns of road use in a Chihuahuan Desert snake community. To do so, I conducted standardized road-cruising surveys in southern New Mexico where I measured all alive and dead snakes encountered, which revealed species-specific, seasonal, and demographic patterns in the frequency of road use and mortality. In the most commonly encountered species, I found evidence of directional selection against larger body sizes from vehicle collisions, albeit from a small sample, and pilot simulations based on these data indicated that evolutionary change is one possible result of such non-random mortality. I note that this was an observational study that lacked a true control population not subjected to road-induced mortality. Regardless, this study highlights the vulnerability of snake communities to road-related threats overall and also shows how within-species patterns of road mortality may influence populations through the selective removal of larger individuals. My results emphasize the need to develop targeted mitigation measures as road networks continue to expand globally, especially for larger snakes with high levels of road use that may be more prone to vehicle collisions.

Methods

From April to November in 2017, I systematically surveyed snakes starting shortly after sunset by driving along a 37-km stretch of road in southern Doña Ana County, New Mexico, USA. With the high beams on in a 2012 Nissan Altima, I visually inspected both lanes of the road for snakes while driving at approximately 55 km/h (Fitch, 1987). I began the survey just north of Chaparral on New Mexico State Road 213 (i.e., War Road), which is a two-lane, asphalt-paved road, and I drove until reaching the gate of White Sands Missile Range, where I turned around and continued surveying the road back towards Chaparral for a total distance of 74 km per visit. I conducted three surveys per month, an average of 8 days apart, for a total of 24 surveys. The average time that the first snake was detected across surveys was 2,040 h, while the earliest time of first snake detection was 1,852 h. The average time that the last snake was detected across surveys was 2,200 h, while the latest time of last snake detection was 2,331 h. During 2014–2016, opportunistic surveys were conducted along the same stretch of road in the same manner described above: seven surveys in 2014 (three in April, one in May, and three in July), six in 2015 (four in September and two in October), and three in 2016 (two in June and one in September). Data from opportunistic surveys were analyzed with those from standardized surveys for body sizes, proportion of DORs, and other community-level analyses, but only standardized data were used for aspects of seasonal activity and spatial hotspots. This work was conducted under New Mexico Department of Game and Fish (permit # 1329).

Alive-on-road (AOR) and dead-on-road (DOR) snakes were treated the same when encountered. I recorded the following data upon capture of a snake: Species identification, DOR status (yes or no), time, GPS location, sex determined by caudal palpation, body size as snout-vent length (SVL) and tail length (TL) measured to the nearest 0.1 cm with a tape measure, and weight to the nearest 1 g with an electronic scale. For recognition of individual snakes, I applied three marks with a cauterizer (Taxonyx Science Inc., El Monte, CA, USA) to the underside of their tail (Ferner, 2007). Alive snakes were processed and released on the other side of the road. Dead snakes were processed and moved into natural habitat away from the road. Some snake carcasses were not measured for all traits because their advanced state of morbidity would have produced distorted morphometrics, such as measuring length but not weight. A handful of other snakes were identified and recorded but traffic patterns at the time of encounter prohibited their capture for measurements.

I used Program R version 4.4.3 (R Core Team, 2025) with the R studio interface (Posit Studio, 2025) to analyze data, which was organized in Microsoft Excel (Redmond, WA, USA). I assessed seasonal activity by plotting the number of snakes observed per month by species from standardized surveys. To determine if there were any spatial hotspots of road mortality in standardized surveys, I mapped the position of DOR and non-DOR snakes using the R package ggmap (Kahle & Wickham, 2013). To determine if snake encounters were evenly distributed along the road, I binned snake locations into 2-km intervals and performed Chi-squared tests on the resulting histograms comparing actual to expected distributions. I used data from standardized surveys to compare with results collected in an analogous manner by similar road-cruising survey also conducted in Doña Ana County, New Mexico (Price & LaPointe, 1990), which occurred during 1975–1978 along Interstate 25 that is about 75 km from State Road 213 (i.e., War Road). Using data from my standardized and opportunistic surveys, I examined the ranges of body sizes between DOR and AOR snakes to determine if road-killed snakes were representative of the populations at large. I also explored body size variation within the most frequently encountered species with boxplots and two-sample t-tests between adult males and females to understand if my samples were representative of the species in the region (Degenhardt, Painter & Price, 1996).

I used data for the most frequently encountered species to determine if body size variation was related to road mortality. Data for Western Diamondback Rattlesnakes (Crotalus atrox) were pooled across all years. Overall, 91 C. atrox were observed during this study, 20 of which were observed but not measured. The final dataset included 48 individuals based on each snake having all three measurements of body size (SVL, TL, and weight) and were most likely to be in their second year of life with an SVL >40 cm (Campbell & Lamar, 2004), thus were not neonates which are presumably road naïve. I fit binomial logistic regression models that related DOR status (yes or no) to body size as SVL or total length (SVL plus tail length (not including length of rattle segments)) with fixed factors for weight, sex, year, and month, and combinations of these factors as interactions. Using the Akaike Information Criterion (AIC), I compared 11 models to determine the best fit model to the data (lowest AIC value). For comparison, I also fit the same models to a larger dataset that included C. atrox with SVL <40 cm (n = 57 snakes). Using the top model, I calculated odds ratios and 95% confidence intervals (CI) to evaluate the impact of body size on probability of road mortality. Given than males of C. atrox are on average larger (Beaupre, Duvall & O’Leile, 1998) and exhibit greater movement (DeSantis et al., 2019) than females, I further explored the model with sex as an interaction with size to assess the covariation of body size and road use. Lastly, I calculated the area under the curve (AUC) for the top models to determine the model’s ability to discriminate between groups (AOR vs. DOR).

To visualize selection on the significant variable from the top binomial model in C. atrox, I estimated a nonparametric cubic spline for total length (Schluter, 1988) and plotted a selection surface for road mortality against body condition (total length vs. weight) (Schluter & Nychka, 1994). I used prediction probabilities from the top binomial model to calculate the cubic spline and selection surface. To further understand how the size of a snake relates to its likelihood of collision with a vehicle, I used a Monte Carlo approach to simulate 10,000 road crossings on a two-lane highway, accounting for realistic vehicle constraints and body sizes. The probabilistic model assumed that snakes crossed at a random position along the road’s width (732 cm wide), which was based on measurements from Google Earth for State Road 213. Cars in the simulations had two wheels set at 20 cm wide each (standard width for most car wheels in the United States), which were fixed at 177 cm apart from each other (standard width for most cars in the United States). Snakes were assigned body lengths based on empirical distributions from DOR and non-DOR individuals (means and standard errors), with the range constrained by the maximum total length I observed (135 cm). I modeled two scenarios: (1) a single vehicle passing, and (2) two vehicles passing simultaneously. A snake was deemed dead if any part of a wheel overlapped with its body. I visualized calculated mortality probabilities for different sizes using a smoothed probability curve with 95% CIs.

To explore the possible evolutionary implications of size-based mortality for C. atrox, I used total length differences between DORs and AORs to estimate the standardized selection differential (s) and its standard deviation (SD) using 1,000 bootstrap replicants (Price et al., 1984). Data were standardized using the scale function in The R Base Package (v. 4.5.0). To explore directional selection in C. atrox, I simulated total length change in 1,000 populations over 50 generations using the Breeder’s equation (Lush, 1937): Z = h2*s, where Z is the response to selection, s is the selection differential, and h2 is heritability. I used the estimate for s and its SD for C. atrox measured herein, and an h2 estimate and its SD for body size measured from wild populations of the pitviper, Gloydius blomhoffii (h2 = 0.59, SD = 0.27: Sasaki, Fox & Duvall, 2009). All simulated populations started at the mean total length that I measured for adult C. atrox (80.66 cm). To account for the possibility that populations acclimate to road mortality, I included a decelerating effect in the model where selection weakened over time (decay coefficient, k = 0.005).

Results

Seasonal activity and road use

From 24 standardized surveys during 2017 covering 1,776 km of road, I encountered 101 snakes representing 10 species (Fig. 1; Table 1). Seasonal activity peaked in September (25 snakes) and was lowest in November (three snakes), while the most species were detected in August (seven species) and the lowest in November (two species) (Fig. 2). I detected 4.2 snakes per visit, which ranged from a low of 1 per visit in November to a high of 8.3 per visit in September. Overall snake density was 0.057 snakes per km and followed the same seasonal pattern. Most encounters were of Crotalus atrox (47.5% of observations) with a peak in August (n = 13), and Arizona elegans (13.9% of observations) with a peak in May (n = 6). From 16 opportunistic surveys during 2014–2016 covering 1,184 km of road, I encountered 91 snakes (5.7 snakes per visit and 0.077 snakes per km) representing eight species. Most of the opportunistic encounters were of C. atrox (48.4% of observations) and C. viridis (18.7% of observations). A comparison between standardized surveys and results from Price & LaPointe (1990) revealed similar patterns but several notable differences (Table 2). For example, that study detected more species and more snakes per km but had fewer snakes per visit. Both studies had a similar proportion of their community made up by observations of C. atrox (almost 50% in each), but the second most common species differed. Peak months for snake numbers and species diversity also differed between studies but were within one month.

Figure 1 Snake community in the Chihuahuan Desert.

Snake species found during road-cruising surveys near Chaparral, Doña Ana County, New Mexico. (A) Western Diamondback Rattlesnake (Crotalus atrox); (B) Glossy Snake (Arizona elegans); (C) Prairie Rattlesnake (Crotalus viridis); (D) Sonoran Gophersnake (Pituophis catenifer); (E) Long-nosed Snake (Rhinocheilus lecontei) (F) Chihuahuan Nightsnake (Hypsiglena jani); (G) Plains Black-headed Snake (Tantilla nigriceps); (H) Desert Kingsnake (Lampropeltis splendida); (I) Coachwhip (Masticophis flagellum); and (J) Big Bend Patchnose Snake (Salvadora deserticola). All photos taken by DFH.

Table 1 Snake encounters in the Chihuahuan Desert.

Summary of snake encounters from road-cruising surveys conducted in 2017 along a 37 km stretch of road near Chaparral, Doña Ana County, New Mexico. AOR = alive-on-road; DOR = dead-on-road.

	April	May	June	July	August	September	October	November		
	AOR	DOR	AOR	DOR	AOR	DOR	AOR	DOR	AOR	DOR	AOR	DOR	AOR	DOR	AOR	DOR	Total	
Arizona elegans	2	–	4	1	1	1	–	–	1	1	1	2	–	–	–	–	14	
Crotalus atrox	5	–	–	1	5	1	1	1	9	4	8	4	3	4	1	1	48	
Crotalus viridis	1	–	–	–	–	–	–	1	2	–	3	1	1	1	–	–	10	
Hypsiglena jani	–	–	–	–	1	–	–	–	2	–	–	–	–	–	–	1	4	
Lampropeltis splendida	–	–	–	–	–	–	1	–	–	–	–	–	–	–	–	–	1	
Masticophis flagellum	–	–	–	–	–	–	–	–	–	1	–	–	–	–	–	–	1	
Pituophis catenifer	1	–	–	–	–	–	–	–	1	–	1	3	2	2	–	–	10	
Rhinocheilus lecontei	1	1	1	1	1	1	1	–	1	–	–	1	1	–	–	–	10	
Salvadora deserticola	–	–	–	–	–	–	–	–	–	–	–	1	–	–	–	–	1	
Tantilla nigriceps	–	–	–	–	1	–	1	–	–	–	–	–	–	–	–	–	2	
Total snakes	11	8	12	6	22	25	14	3	101	
Total species	5	3	5	5	7	6	4	2	10	
AOR: DOR	10	1.7	3	2	2.7	1.1	1	0.5	1.9	
% DOR	9	38	25	33	27	48	50	67	36	
Snakes per km	0.050	0.036	0.054	0.027	0.099	0.113	0.063	0.014	0.057	
Snakes per visit	3.7	2.7	4	2	7.3	8.3	4.7	1.0	4.2	

Figure 2 Snake activity patterns.

Seasonal activity of snake species found during road-cruising surveys in 2017 conducted three times per month along a 37 km stretch of road near Chaparral, Doña Ana County, New Mexico.

Table 2 Comparison to historical study.

Comparison between standardized surveys in this study and a historical study conducted on a road about 75 km away also in Doña Ana County, New Mexico. *Values represent 2017 data from New Mexico Department of Transportation.

	This study	Price & LaPointe (1990)	
Species	10	18	
Snakes per km	0.057	0.071	
Snakes per survey	4.21	1.79	
AOR: DOR	1.86	1.28	
% DOR	35.6	43.8	
% Crotalus atrox	47.5	46.0	
% of Crotalus atrox DOR	31.9	45.5	
Second most common species	Arizona elegans	Pituophis catenifer	
Peak month for snake activity	September	August	
Peak month for snake diversity	August	June/July	
km travelled	1,776	6,350	
Surveys per month	3	13–61	
Year surveyed	2017	1975–1978	
Average annual daily traffic*	3,000–7,999 cars	8,000–14,999 cars	

Community patterns of road mortality

Spatially, standardized surveys revealed that both AOR (Fig. 3A) and DOR (Fig. 3B) snakes were about equally distributed across the 37-km stretch of road. Grouping snake observations into 2-km bins revealed that there were no areas along the road that exhibited elevated numbers of snakes found AOR (Fig. 3C) or DOR (Fig. 3D). The percentage of DOR snakes differed between standardized surveys and those of Price & LaPointe (1990), which was conducted on a road with higher traffic (Table 2). Data from both standardized and opportunistic surveys revealed similar rates of road mortality relative to each dataset individually. For example, using standardized encounters, 35.6% of snakes were found DOR with a ratio of 1.9 AOR: 1 DOR, and 33% were found DOR with a ratio of 1.8 AOR: 1 DOR in opportunistic encounters. The most frequently encountered species in both datasets, C. atrox, exhibited similar DOR proportions: 33% of C. atrox were found DOR in standardized surveys and 31.8% in opportunistic surveys. Combined, 34% of all snakes were found DOR, a proportion that is comparable to rates within most species and sexes (Table 3). Across all species, males and females exhibited similar DOR proportions at 37% and 40%, respectively, while juveniles had a DOR rate of 17%. Individuals of the two diurnal species (Masticophis flagellum and Salvadora hexalepis) were all found DOR.

Figure 3 Spatial distribution of snake encounters.

Maps of snake encounters during road-cruising surveys in 2017 conducted three times per month along a 37 km stretch of road near Chaparral, Doña Ana County, New Mexico. Spatial distribution of snakes found alive-on-road (A, C) and dead-on-road (B, D). See methods for more details. Images modified from Google Earth (Image TerraMetrics).

Table 3 Morphological summary of snake species.

Morphological and mortality data for snakes encountered during road-cruising surveys conducted during 2014–2017 along a 37 km stretch road near Chaparral, Doña Ana County, New Mexico. Means are presented ± standard deviations with ranges in parentheses.

Species	Sex	n	SVL (cm)	TL (cm)	Weight (g)	% DOR	
Arizona elegans	M	8	33.51 ± 10.26 (21.7–54.2)	6.15 ± 1.81 (3.6–9.6)	12.33 ± 7.39 (3–22)	37.5	
F	5	62.84 ± 7.06 (55.2–69.6)	11.80 ± 1.34 (10.3–13.8)	112.50 ± 32.77 (69–147)	0	
J	7	26.44 ± 5.35 (20.4–33.2)	4.31 ± 1.04 (3.1–5.8)	7.86 ± 5.27 (3–18)	14.3	
Unknown	2	–	–	–	100	
Overall	22	38.37 ± 16.68 (20.4–69.6)	6.92 ± 3.31 (3.1–13.8)	34.06 ± 47.38 (3–147)	27.3	
Crotalus atrox	M	47	78.86 ± 21.20 (40.6–123)	7.05 ± 1.89 (3.2–11.7)	391.66 ± 363.70 (42–1511)	36.2	
F	17	68.76 ± 14.29 (43.5–86.5)	4.70 ± 0.96 (3–6.3)	243.69 ± 151.27 (38–505)	35.3	
J	21	32.66 ± 3.02 (29.1–39.5)	2.83 ± 0.54 (2.2–4.1)	19.89 ± 4.57 (15–28)	14.3	
Unknown	6	–	–	–	50	
Overall	91	68.78 ± 24.51 (29.1–123)	5.81 ± 2.27 (2.2–11.7)	291.42 ± 312.41 (15–1511)	31.9	
Crotalus viridis	M	14	66.39 ± 15.38 (37–93.5)	5.99 ± 1.45 (2.7–8.2)	205.50 ± 149.69 (25–556)	7.1	
F	6	68.90 ± 7.14 (62.5–8)	4.54 ± 0.82 (3.7–5.6)	174.50 ± 70 (125–224)	66.7	
J	5	30.65 ± 0.49 (30.3–31)	2.30 ± 0 (2.3–2.3)	13.50 ± 7.78 (8–19)	20	
Unknown	2	–	–	–	0	
Overall	27	63.29 ± 17.01 (30.3–93.5)	5.22 ± 1.70 (2.3–8.2)	173.64 ± 143.60 (8–556)	22.2	
Hypsiglena jani	M	5	29.56 ± 5.27 (24.9–37.5)	5.46 ± 0.74 (4.7–6.3)	16 ± 4.24 (13–19)	0	
F	3	36.70 ± 7.37 (28.4–42.5)	5.27 ± 1.01 (4.1–5.9)	15.50 ± 10.61 (8–23)	66.7	
Overall	8	32.24 ± 6.71 (24.9–42.5)	5.39 ± 0.79 (4.1–6.3)	15.75 ± 6.60 (8–23)	25	
Lampropeltis splendida	M	1	72.5	11	124	0	
F	1	82	12.8	–	100	
J	2	32.55 ± 1.34 (31.6–33.5)	6.30 ± 2.12 (4.8–7.8)	11 ± 1.41 (1–12)	0	
Overall	4	54.90 ± 26.11 (31.6–82)	9.10 ± 3.53 (4.8–12.8)	48.67 ± 65.25 (1–124)	25	
Masticophis flagellum	M	2	111	34.5	–	100	
Unknown	2	–	–	–	100	
Pituophis catenifer	M	8	54.86 ± 24.67 (37.9–98.4)	8.77 ± 3.78 (5.8–14.9)	23 ± 5.10 (16–3)	50	
F	9	51.31 ± 20.97 (39.7–106.5)	7.18 ± 2.17 (5.5–12.8)	74.88 ± 128.66 (23–393)	44.4	
Overall	17	52.86 ± 21.94 (37.9–106.5)	7.88 ± 2.98 (5.5–14.9)	54.92 ± 101.75 (16–393)	47.1	
Rhinocheilus lecontei	M	11	48.85 ± 13.92 (24.1–66.2)	8.18 ± 2.37 (3.9–11)	43.57 ± 35.88 (5–102)	63.6	
F	1	71.3	10.8	110	0	
J	3	22.53 ± 1.01 (21.6–23.6)	3.73 ± 0.55 (3.2–4.3)	5.33 ± 1.53 (4–7)	33.3	
Overall	15	45.08 ± 17.54 (21.6–71.3)	7.47 ± 2.87 (3.2–11)	39.18 ± 40.40 (4–11)	53.3	
Salvadora deserticola	M	1	52.3	14.1	34	100	
Tantilla nigriceps	F	2	19.65 ± 0.07 (19.6–19.7)	4.05 ± 0.07 (4–4.1)	2.50 ± 0.71 (2–3)	0	

Snake body sizes

The range of body sizes for all DOR snakes with measurements was nearly identical to the range of body sizes for all AOR snakes (DOR SVL range = 22.1–120 cm, n = 50; AOR SVL range = 19.6–123 cm, n = 107). The same pattern of overlapping body size ranges between DOR and AOR snakes was also found in the five most frequently encountered species: Crotalus atrox (DOR SVL range: 30.2–120 cm, n = 21; AOR SVL range: 29.1–123 cm, n = 50), Rhinocheilus lecontei (DOR SVL range = 23.6–66.2 cm, n = 8; AOR SVL range = 21.6–71.3 cm, n = 7), Pituophis catenifer (DOR SVL range = 37.9–98.4 cm, n = 7; AOR SVL range = 38.4–106.5 cm, n = 9), and Crotalus viridis (DOR SVL range = 30.3–80.0 cm, n = 5; AOR SVL range = 31.0–93.5 cm, n = 14). In the fifth species, Arizona elegans, the range of DOR body sizes was more limited than the range of AOR body sizes (DOR SVL range = 22.1–34.6 cm, n = 4; AOR SVL range = 20.4–69.6 cm, n = 16), likely due to a small sample size. Using measurements from both AOR and DOR snakes for these five species that possessed adequate samples for comparison, I found that morphometrics generally conformed to known sex-based differences within each species for SVL, TL, and weight (Fig. 4; Table 3). Together with the overlapping body-size ranges, these results indicate that the segment of the populations I sampled were representative of the body sizes for these species within the area.

Figure 4 Morphological summary of common snake species.

Box plots showing body size metrics for the five most encountered snake species from road-cruising surveys conducted during 2014–2017 along a 37 km stretch of road near Chaparral, Doña Ana County, New Mexico. Measurements come from alive and dead snakes. Note that sample sizes may differ among traits. *Indicates significant difference in a two-sample t-test at P < 0.05.

Size-based mortality in Crotalus atrox

A comparison of binomial logistic regression models found that total length was a significant predictor of road mortality in analyses of a dataset excluding snakes with SVL <40 cm, and another with individuals of all sizes, where the top models (based on lowest AIC values) were the same for both datasets and included the factors of total length, weight, and sex (Tables 4, 5). The following results are for the dataset that included C. atrox with SVL >40 cm. Using the top model, which had an AUC of 0.7518, I found that the total length increased the odds of road mortality at a ratio of 1.095 ([1.011–1.212] 95% CI). Essentially, for every 1 cm increase in total length, the probability of road mortality increased by 9.5% ([1.1–21.2%] 95% CI). Confidence intervals for the factors sex (odds ratio = 0.312, [0.059–1.521] 95% CI) and weight (odds ratio = 0.996, [0.989–1.001] 95% CI) overlapped with one. With the smallest snakes included in the model (AUC = 0.7542), the overall odds ratio for total length increased slightly to 1.101 ([1.017–1.218] 95% CI) but wide confidence intervals for sex and weight still overlapped with one. The mean total length of adult C. atrox found DOR (88.46 ± 17.07 cm SD, range = 53.3–131.7 cm, n = 18) was nearly 10 cm larger than the mean of those found AOR (79.93 ± 23.26 cm SD, range = 43.8–133.9 cm, n = 41), and the same pattern was present when using a dataset that included all C. atrox with length measurements (mean DOR total length = 81.14 ± 24.25 cm SD, range = 32.7–131.7 cm, n = 21; mean AOR total length = 71.83 ± 27.35 cm SD, range = 31.9–133.9 cm, n = 50).

Table 4 Models for factors that affect road mortality.

Model comparisons for factors influencing dead-on-road (DOR) status in the Western Diamondback Rattlesnake (Crotalus atrox) found on a 37 km stretch of road during 2014–2017 in Doña Ana County, New Mexico. Models ranked in descending order by lowest Akaike Information Criterion (AIC) values based on. Bold text indicates top model for each dataset.

Model	df	AIC	
Only Crotalus atrox >40 cm in SVL (n = 48)	
DOR ~ Total length + Weight + Sex	4	53.28896	
DOR ~ SVL + Weight + Sex	4	53.57903	
DOR ~ Total Length * Sex + Weight	5	53.91065	
DOR ~ Total Length + Sex	3	54.25181	
DOR ~ SVL * Sex + Weight	5	54.26822	
DOR ~ SVL + Sex	3	54.32172	
DOR ~ Weight * Sex + SVL	5	55.27788	
DOR ~ Total Length + Weight + Sex + Month + Year	12	58.25513	
DOR ~ SVL + Weight + Sex + Month + Year	12	58.65616	
DOR ~ Total Length + Weight + Sex + Month	11	59.31772	
DOR ~ SVL + Weight + Sex + Month	11	59.64136	
All Crotalus atrox (n = 57)	
DOR ~ Total Length + Weight + Sex	5	65.99735	
DOR ~ SVL + Weight + Sex	5	66.40891	
DOR ~ Total Length + Sex	4	67.43438	
DOR ~ SVL + Sex	4	67.54470	
DOR ~ Total Length * Sex + Weight	7	68.35017	
DOR ~ SVL * Sex + Weight	7	68.85360	
DOR ~ Weight * Sex + SVL	7	70.03038	
DOR ~ Total Length + Weight + Sex + Month + Year	13	70.75077	
DOR ~ SVL + Weight + Sex + Month + Year	13	71.34855	
DOR ~ Total Length + Weight + Sex + Month	12	73.46662	
DOR ~ SVL + Weight + Sex + Month	12	73.87667	

Table 5 Top models for factors affecting road mortality.

Results from the top models for factors influencing road mortality in the Western Diamondback Rattlesnake (Crotalus atrox) found on a 37 km stretch of road during 2014–2017 in southern New Mexico. Bold text indicates significance in each dataset.

	Estimate	SE	Z	P-value	
Only Crotalus atrox >40 cm in SVL (n = 48)	
Intercept	−6.4924	2.8755	−2.258	0.0240	
Total length	0.0908	0.0450	2.016	0.0438	
Weight	−0.0045	0.0030	−1.504	0.1326	
Male	−1.1640	0.8234	−1.414	0.1575	
All Crotalus atrox (n = 57)	
Intercept	−6.8484	2.8782	−2.379	0.0173	
Total length	0.0966	0.0450	2.144	0.0320	
Weight	−0.0048	0.0030	−1.601	0.1094	
Juvenile	2.8061	1.5830	1.773	0.0763	
Male	−1.1958	0.8294	−1.442	0.1494	

For C. atrox with SVL >40 cm, the model including sex as an interaction with total length was within one AIC value of the top model with a slightly higher AUC (0.7789), suggesting that size effects may differ by sex (Table 4). With sex as an interaction, total length was now marginally significant (estimate = 0.146, SE = 0.077, Z = 1.89, P = 0.059), weight and sex remained non-significant (Table 5), but the interaction between sex and total length was not significant (estimate = −0.077, SE = 0.074, Z = −1.04, P = 0.298). For the dataset with all C. atrox, the model with sex as an interaction with total length was >2 AIC values from the top model, and the pattern of significance for each of the factors was the same as the adult-only dataset. In females, the mean total length of DORs (83.7 ± 3.40 cm SD, range = 79.8–87.6 cm, n = 5) and mean probability of DOR (44.1 ± 14.7% SD, range = 31.7–68.9%, n = 5) were significantly greater than the means of AORs (mean total length = 68.81 ± 16.19 cm SD, range = 47.3–91.7 cm; mean DOR probability = 25.4 ± 14.2% SD, range = 7.7–50.5%, n = 11) (Welch’s t = 2.17, df = 12, P = 0.013; Welch’s t = 2.31, df = 8, P = 0.044). In males, the mean total length of DORs (90.98 ± 20.69 cm SD, range = 53.3–112.6 cm, n = 6) and mean probability of DOR (25.6 ± 15.0% SD, range = 3.6–43.4%, n = 6) were greater, but not significantly more, than the means of AORs (mean total length = 82.72 ± 24.36 cm SD, range = 43.8–133.9 cm, n = 26; mean DOR probability = 17.2 ± 13.4% SD, range = 1.8–53.9%, n = 26) (Welch’s t = 2.26, df = 9, P = 0.416; Welch’s t = 2.36, df = 7, P = 0.245).

Using predicted values from the top model for adults only, I found that the probability of road mortality increased with increasing total length, but with wide confidence intervals (Fig. 5A). The selection surface revealed an increased probability of road mortality by body condition with the highest probabilities for snakes around 80–100 cm in total length and 250–350 g in weight (Fig. 5B). The cubic spline demonstrated that DOR probability peaked at nearly 50% for snakes at a total length of around 90–100 cm with a slight decline thereafter and wide confidence intervals (Fig. 5C). From 10,000 Monte Carlo simulations, the probability of road mortality increased with total length in both single-car and two-car scenarios (Fig. 5D). The mean probability of road mortality in the single car scenario (25.8%, SD = 43.7%, [24.9–26.6%] 95% CI) was lower than the mean in the two-car scenario (40.6%, SD = 49.1%, [39.6–41.5%] 95% CI), with non-overlapping confidence intervals.

Figure 5 Road mortality patterns in Crotalus atrox.

Size-based mortality in adult Western Diamondback Rattlesnakes (Crotalus atrox) found along a road near Chaparral, Doña Ana County, New Mexico. (A) Probability of an individual being found dead-on-road (DOR) in relation to total length with 95% confidence interval in grey. (B) Selection surface for body condition (total length by weight) with color of individuals indicative of the probability of being found DOR. (C) Cubic spline showing the probability of an individual being found DOR by total length with 95% confidence interval in blue. (D) Probability of an individual being found DOR based on Monte Carlo simulations of snakes crossing a two-lane highway, accounting for vehicle dimensions and snake body sizes, for one-car and two-car scenarios with 95% confidence intervals in grey.

Simulating potential evolutionary change in Crotalus atrox

I estimated s to be −0.421 (± 0.268 SD) for total length differences between adult DOR and AOR C. atrox. Using s and the h2 estimate from Sasaki, Fox & Duvall (2009) (h2 = 0.59, SD = 0.27), I simulated evolutionary change in 1,000 populations over 50 generations, which revealed that total length decreased from the mean size in most simulations (Fig. 6A). The global mean total length in all simulated populations after 50 generations was 68.8 cm ([68.2–69.4] cm 95% CI) (Fig. 6B). The simulated populations that decreased in size ended with a mean total length of 67.8 cm ([67.17–68.34 cm] 95% CI), whereas populations that increased in size ended with a mean total length of 84.1 cm ([83.31–84.85 cm] 95% CI). The relationship between heritability and final total length in simulations was significant and negative, where higher estimates of h2, on average, produced more significant reductions in total length (Fig. 6C).

Figure 6 Models of evolutionary change in Crotalus atrox.

Simulations of morphological change in the Western Diamondback Rattlesnake (Crotalus atrox). Input data for models were derived from adult C. atrox found dead-on-road and alive-on-road near Chaparral, Doña Ana County, New Mexico. (A) Change in total length of 1,000 populations over 50 generations depicting whether they were increasing or decreasing from the mean size of adults in this population (80.66 cm). (B) Average change in total length from all simulated populations (black line) with 95% confidence interval in grey. (C) Linear relationship between the final total length in simulated populations and heritability (h2).

Discussion

The results of this study suggest that roads may act as both ecological traps and selective filters for snakes (Rosen & Lowe, 1994; Shine et al., 2004), particularly in species such as C. atrox which can attain large sizes and exhibit high levels of road use. I detected seasonal variation in road use, with a peak in late summer to early fall for the entire snake community, a common finding among road-cruising studies in the Chihuahuan Desert that is associated with the wet monsoon period (i.e., Campbell, 1953; Pough, 1966; Reynolds, 1982; Reynolds & Scott, 1982; Price & LaPointe, 1990; Degenhardt, Painter & Price, 1996; Pendley, 2001). Notably, I detected no distribution patterns of snakes found dead-on-road (DOR) or alive-on-road (AOR), which is in contrast to other studies that detected spatial hotspots, often linked to distinct habitat features along roads (e.g., Langen, Ogden & Schwarting, 2009; Garrah et al., 2015; Wagner, Brune & Popescu, 2021). The snake community I documented was depauperate in terms of overall diversity relative to a historical road-cruising study conducted only about 75 km away in New Mexico (10 vs. 18 snake species) (Price & LaPointe, 1990), suggesting that the habitat was homogeneous along the road I surveyed (Bateman & Merritt, 2020), which may have contributed to the lack of spatial hotspots. Nevertheless, the proportion of one species, the Western Diamondback Rattlesnake (Crotalus atrox), represented nearly 50% of all snake encounters in both studies with comparable, albeit elevated, rates of DOR in the historical study (Price & LaPointe, 1990). All of the other species I documented were also found by Price & LaPointe (1990), by Mendelson & Jennings (1992) from a road in southwestern New Mexico about 245 km away who found 23 total species, and by Reynolds (1982) from a road in Mexico about 330 km away who found 20 total species. Overall, road mortality rates were higher in the study by Price & LaPointe (1990) relative to what I detected, which was likely due to higher vehicle traffic on the road that they surveyed (Traffic Monitoring Program, New Mexico Department of Transportation (www.dot.nm.gov)). Still, the variation in road mortality between species, sexes, and size classes I found suggests that vehicle collisions may not be random (Shine et al., 2001). In particular, I found that road mortality was linked to body size in C. atrox, and pilot simulations based on these data suggested that vehicle collisions have the potential to induce a change in mean total length in this population over time. My preliminary findings and exploratory projections lend support to a growing body of research showing that humans can exert direct pressure on wildlife populations through unnatural selection, which can cause evolutionary changes in species (Allendorf & Hard, 2009; Sullivan, Bird & Perry, 2017; Grzegorczyk et al., 2024).

There are several factors that should be considered when interpretating these results, such as carcass persistence because animals are often removed by scavengers after they are hit by a car, or their bodies are destroyed by continued vehicle collisions (Boyle et al., 2025). It could be that smaller C. atrox that experienced road mortality were removed by scavengers or degraded faster than larger individuals. Studies that have examined carcass persistence in snakes have found that most individuals were removed within 24 h (some persist for up to 1 week: Antworth, Pike & Stevens, 2005; Santos, Carvalho & Mira, 2011) with variation across ecosystems (about 60 h for most to disappear in arid areas: Hubbard & Chalfoun, 2012), but there have been mixed results for the relationship between snake size and carcass persistence (Degregorio et al., 2011; Cabrera-Casas, Robayo-Palacio & Vargas-Salinas, 2020). While carcass persistence was not explicitly measured in this study, I think that the disappearance of smaller individuals had a negligible impact on the results because DOR snakes exhibited a nearly identical range of body sizes compared to AOR snakes overall, including within C. atrox and in several of the smaller-sized species (e.g., A. elegans, C. viridis, P. catenifer and R. lecontei), indicating that the body sizes of carcasses measured in this study were not systematically biased towards larger individuals. Regardless, I only conducted 40 total surveys and carcass persistence was not measured, thus it is clear that the road-mortality rate I detected is a vast underestimation of the true frequency of vehicle collisions with snakes along this road. Another consideration is disentangling road mortality from other selective pressures acting on body size. While simulations suggested a decrease in mean total length over 50 generations, other environmental factors, such as changes in climate, habitat, and prey were not modelled. These factors will play roles in shaping body sizes over time (e.g., Madsen & Shine, 2000), especially in C. atrox (Amarello et al., 2010), which could offset or even reverse the trend I projected. Additionally, behavioral differences among individuals, such as boldness or road avoidance (Andrews & Gibbons, 2005), could influence the trajectory of evolutionary change. Studies that incorporate animal movements (Bénard, Lengagne & Bonenfant, 2024), habitat-level assessments (Wagner, Brune & Popescu, 2021), and experimental approaches to evaluate whether individuals that avoid roads differ systematically from those that do not (Shine et al., 2004) will be useful. Explorations into whether learned avoidance behaviors can mitigate the evolutionary impacts of roads would also be fruitful. Lastly, the evolutionary models I constructed were based on a heritability estimate for neonatal body length from a different pitviper species (Sasaki, Fox & Duvall, 2009), which appears to be the only published estimate of this trait for any snake species (Postma, 2014). An estimate of heritability for body size in C. atrox from this population and monitoring DORs over a longer period of time to capture generation-to-generation changes in body size at this site would improve the precision of these models. Ultimately, more road ecology research incorporating evolutionary data from this site and others will serve as a test of the road-induced selection hypothesis, which I propose accounts for the trend in C. atrox documented herein.

Several studies have shown that drivers intentionally target snakes on roads when compared to random objects (Ashley, Kosloski & Petrie, 2007; Beckmann & Shine, 2012) and interviews with drivers reveal that they will deliberately hit snakes more often than other animals (Langley, Lipps & Theis, 1989). For large, conspicuous snakes crossing roads at night, this likely means that they are targeted at higher rates than small, cryptic snakes. Several traits possessed by C. atrox increase its conspicuousness on roads, such as a large body size (both in terms of length and girth), a distinct black-and-white banded tail, and the propensity for individuals to lift their entire rattle into the air while moving across roads (D.F. Hughes, 2017, personal observation). This combination of traits may predispose this species to higher rates of targeting by drivers, especially if drivers recognize that it is a venomous species (Kontsiotis, Rapti & Liordos, 2022), which are persecuted much more frequently than non-venomous species (Larson et al., 2024). Larger snakes are not only easier to see for drivers that are attempting to intentionally run them over, but they are also harder to miss for drivers endeavoring to avoid the snake. From DOR C. atrox that I could identify the point of impact from a wheel (n = 16), most were hit at midbody (56%), second most were in the head (31%), and the rest were on the tail (13%). Running over a large snake at midbody suggests that drivers either intentionally targeted the snake, failed to see the snake in time to avoid it, or could not avoid the snake because another car was passing at the same time (or would have required a potentially dangerous/illegal maneuver to avoid the snake such as veering into the oncoming lane or off the road entirely). Road-crossing simulations using standardized sizes for wheel width, distance between wheels, and width of a two-lane road showed that mortality was highest when two cars passed simultaneously, where the DOR probability approached 80% for the largest snakes measured in this study. In two-car scenarios, vehicle collisions with snakes can only be avoided if they can either pass between the wheels of a car (known as “straddling”), pass between the cars in the middle of the road, or be far enough to the side of the road that a car can avoid them without swerving too much. All of these instances would likely force drivers to make a conscious decision to avoid hitting the snake, which is easier to do if the snake is small (and also does not move). Ultimately, roads have created a novel selection landscape that has unintentionally hurled humans and snakes on an evolutionary collision course, where seemingly minor decisions by drivers behind the wheel may hold great power over the future of another species (Brown & Brown, 2013; Brady & Richardson, 2017; Sullivan, Bird & Perry, 2017; Parlin et al., 2025; Proctor et al., 2025).

The apparent non-random mortality of larger C. atrox on roads raises the possibility that vehicles can exert a form of directional selection on morphology (Brown & Brown, 2013). I note that the cubic spline hinted at disruptive selection where the smallest and largest snakes were among the least likely to exhibit road mortality, but this appears to have been driven by just three individuals with a total length >125 cm found AOR. Such directional selection could contribute to evolutionary change, as predicted by exploratory simulations, which indicated a mean decrease in total length of nearly 12 cm over 50 generations, or approximately 165 years using the average generation time estimated for C. atrox (3.3 years: Castoe, Spencer & Parkinson, 2007). Consequently, roads may hold the potential to be more than just sources of direct mortality—they could represent selective environments capable of shaping evolutionary trajectories (Schell et al., 2021), which can be modulated by driver behaviors (Sullivan, Bird & Perry, 2017). The potential for roads to influence morphological and behavioral traits in vertebrates is not well understood, but recent research has begun to show that vehicle collisions may be more impactful than previously thought (Brady & Richardson, 2017; Sullivan, Bird & Perry, 2017). For example, Parlin et al. (2025) recently found that vehicle collisions (n = 141 DOR squirrels) contributed to the maintenance of a phenotypic cline in coat color for eastern gray squirrels (Sciurus carolinensis) across an urban-rural gradient through selection favoring the melanic morph because it is easier to see on road surfaces in cities (Proctor et al., 2025). Based on 104 road-killed cliff swallows (Hirundo melanogaster) collected over a 30-yr period, Brown & Brown (2013) found that the number of DOR birds decreased despite an increase in overall population size, and that the wing length in the population decreased because selection favored individuals that were more capable of in-air maneuvers needed to avoid vehicles. One study even found that birds with smaller brain sizes were more likely to die by vehicle collisions compared to larger brained birds (Møller & Erritzøe, 2017; but see Wagnon & Brown, 2020). The selection patterns emerging in these recent studies suggest that road mortality may be an important evolutionary factor to consider in road ecology studies (Brady & Richardson, 2017). Conservation strategies should also take these evolutionary effects into account when designing mitigation measures, such as wildlife corridors, road-crossing signs, and road underpasses, to reduce mortality and maintain natural population structures (Van der Grift et al., 2013).

Despite known differences in C. atrox behavior where males move more than females (DeSantis et al., 2019) and inflated male representation in my data, I found that this road-use bias did not confound the relationship between size and DOR status as female DORs were actually the larger of the two sexes, on average, when compared to their AOR counterparts and males followed a similar pattern. Because of this result, I will now explore the possible implications for both trophic ecology and reproduction that could occur from a reduction in body size for C. atrox. Larger C. atrox have larger gape sizes (Hampton & Moon, 2013), thus they eat larger prey (Beavers, 1976) and have higher energetic requirements (Beck, 1995). Smaller body sizes over time could thus alter the current types of prey C. atrox consumes and the frequency at which they consume them (Shine, 1991; Greene & Wiseman, 2023), energetic shifts that would affect predator-prey dynamics with possible implications for human health. For example, C. atrox is a significant predator that can regulate the population sizes of rodent species in the genera Neotoma, Dipodomys, Peromyscus, and Perignathus, which make up the largest proportion of prey in their diet (Beavers, 1976; Reynolds & Scott, 1982; Werler & Dixon, 2000; Campbell & Lamar, 2004). All of these rodent groups were recently identified as reservoirs of Sin Nombre orthohantavirus in New Mexico, which causes disease in humans (Banther-McConnell et al., 2024; Goodfellow et al., 2025). Body size reductions could also influence fitness-linked traits in this species via smaller mothers (Kissner & Weatherhead, 2005). Litter size and body size are often correlated in snakes (e.g., Fitch, 1985), but larger C. atrox mothers do not produce larger litters (Taylor & DeNardo, 2005a; Schuett, Repp & Hoss, 2011), even though they have more yolked ovarian follicles (Tinkle, 1962; Fitch & Pisani, 1993; Rosen & Goldberg, 2002). While smaller mothers may not have reduced litter sizes, other aspects of reproduction could be altered because food intake regulates many traits in this species. For example, compared to wild counterparts, female C. atrox provisioned with supplemental meals reached sexual maturity faster, reproduced more frequently, and had larger offspring (Taylor & Denardo, 2005b; Taylor et al., 2005), suggesting that aspects of female reproduction may change with smaller body sizes. Males could also be impacted by smaller sizes, perhaps, through modified mating tactics. Crotalus atrox exhibits male-biased sexual-size dimorphism (Beaupre, Duvall & O’Leile, 1998) because they conduct ritualized combat for access to mating opportunities (Werler & Dixon, 2000; Campbell & Lamar, 2004). Nevertheless, male body length apparently did not influence the outcome of combat in laboratory trials (Gillingham, Carpenter & Murphy, 1983) and genetic parentage analyses did not find a link between the size of fathers and the number of offspring sired in the wild (Clark et al., 2014). However, larger males were the sole fathers of litters more often, exhibited attendance behavior of females more frequently, and attended pregnant females for longer compared to smaller males (Clark et al., 2014), suggesting that aspects of male reproductive behavior may change with smaller body sizes.

In conclusion, roads may be acting as an agent of selection via non-random mortality by vehicles in a population of C. atrox from southern New Mexico, favoring smaller-bodied individuals over time, which could affect current trophic relationships connected to this large pitviper species. Studies in road ecology have repeatedly shown that larger animals are often more prone to vehicle collisions (e.g., Morelli et al., 2024), but few have collected the data needed to assess the possibility of roads as selective forces (Brady & Richardson, 2017). My data suggest that vehicle collisions are a possible source of morphological change, but I wish to note that my study was observational in nature, was focused on a single species over a short time scale, and, importantly, the evolutionary simulations depended on the assumption that mortality was driven by directional selection on body size as a heritable trait. Consequently, I acknowledge that more data are needed from more sites over longer periods to determine if the trends I found are robust across populations, habitats, and roads. Nevertheless, recognizing roads as drivers of evolutionary change will be important for developing conservation strategies aimed at preserving not just population viability but also the ecological roles and evolutionary potential of species affected by human-modified landscapes (Allendorf & Hard, 2009; Sullivan, Bird & Perry, 2017; Schell et al., 2021; Grzegorczyk et al., 2024).

Supplemental Information

Supplemental Information 1 Raw data of snake ecnounters.

Includes all data from standardized and opportunistic surveys. See Methods for details about data collection.

I dedicate this work to my mentor, collaborator, and friend Walter E. Meshaka Jr., who steadfastly encouraged me with this project and countless others. I also owe a debt of gratitude to Jaclyn M. Adams who helped me during many surveys. I thank four anonymous reviewers for suggestions that improved this manuscript.

Additional Information and Declarations

Competing Interests

Daniel F. Hughes is an Academic Editor for PeerJ.

Author Contributions

Daniel F. Hughes conceived and designed the experiments, performed the experiments, analyzed the data, prepared figures and/or tables, authored or reviewed drafts of the article, and approved the final draft.

Field Study Permissions

The following information was supplied relating to field study approvals (i.e., approving body and any reference numbers):

New Mexico Department of Game and Fish approved this work (permit # 1329).

Data Availability

The following information was supplied regarding data availability:

The raw data are available as a Supplemental File.

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
