# Peer review of "Road ecology of a Chihuahuan Desert snake community: size-based mortality sets the stage for evolutionary change in a widespread pitviper"

_PeerJ, doi:10.7717/peerj.19871_

## Round 0.1 · original submission · Major Revisions

Dear Dr. Hughes, I ask you to respond very carefully to the reviewers' comments. These comments are really well-founded and reasoned. I hope that your responses will help the reviewers make the right decision when reading the new version of your article.

Reviewer 1 ·

Basic reporting

This was an interesting study on road survey encounters of a desert snake community. Although encounter rates, community structure etc are documented a central focus of the study was on the effects of road mortality on the most commonly encountered species.
The paper was well-written and the topic and background of the study were well-described in the introduction. Relevant literature on road surveys of snake communities was cited and the scope and aims of the study were stated.

Experimental design

The research questions were well defined, and the survey methods and analyses were appropriate to address the questions.

Validity of the findings

The implications of road mortalities on rattlesnake populations are fully elaborated. Many potential implications are speculative and attention is paid to relevant caveats.

Additional comments

This was an interesting study that value-added a lot to road survey data on a desert snake community. The questions and focus were well laid out in the introduction and the methods and analyses were well-described and appropriate. The interpretations were reasonable and the potential weaknesses were acknowledged and discussed.

I have a few specific comments/suggestions-

L51. ‘high mortality due to vehicle collisions’ makes it sound like cars hitting each other and killing occupants. Maybe reword to ‘high mortality due to vehicles striking snakes’.

L199. Clarify here and in the heading for Fig 3. That A and C are locations of live snakes and B and D are locations of dead snakes (I think?)

L234. I don't understand the rationale of excluding snakes >40 cm from analyses on the effects of body size on risk of getting run over.

L283. I don't understand the connection in this sentence. Why would a depauperate community suggest that the environment is homogenous?

L332. As an additional caveat here, I think you should mention that the h2 estimate from the Sakai reference was for neonate size. Adult size could have some different genetic architecture.

Fig. 3 Explain in the heading what A, B, C, D panels represent

Reviewer 2 ·

Basic reporting

The language is clear. However, the article reads as a report rather than a research paper. The Results section is very long and hard to follow; the manuscript would benefit from a complete re-writing of the Results to cue on the important findings instead of reporting every single number, and make use of Tables for the many numbers reported. The paper cites relevant literature but the reference list is >100 references long!!! This is not a review, but it makes sense if it was a technical report that was converted to a research article.

Figure 5 and data reported in Results are redundant
Figure 6 A and C present very different answers on for the same question; the author should decide which analysis is 'correct'

Experimental design

There is no experimental design per se. The study presents data from several sources: a systematic survey in 2017, opportunistic surveys a few years earlier, data from the 70's and the 90's. The amalgamation of these data into a coherent story is difficult, and the story, data used for various parts of the analysis etc. is rather confusing.
The data collected in 2017 is suitable for analyses, but trying to integrate data from all these surveys studies into a single analysis is not recommended. As such, it is difficult to determine which data was used for which analyses and why.
I made many comments below highlighting areas where data integration may not be suitable, areas of confusion etc.

Regression analysis is theoretically correct, but the sample sizes are very small for parameterizing complex models and there is no goodness-of-fit test for predicting the probability of death on road. There are many competing top models which say the same thing (within 2 AICc units), but the author only reports he top model (again, with no measure of fit).
I am not an evolutionary biologist, thus I cannot evaluate the last analysis on selection, but I have many questions about the validity of the assumptions, the data used, the logic etc. Also, given that this analysis is based on the logistic regression for DOR vs AOR animals and that regression is questionable due to low sample size and statistical power, the entire analysis on selection is also on very shaky ground.
Some analyses, such as the one vs two car mortality, are disjunct from the rest of the paper.

Overall, the data is great and useful but the story is very complicated, there are many analyses that do not speak to each other, and the selection analysis is quite a stretch given all of the above.

Validity of the findings

Please see many points about statistical analyses at point #2 and as comments in the attached pdf. Overall, the findings about larger snakes getting killed on road is very likely biased by asymmetrical detection probability of DOR (all will be found) vs AOR (imperfect detection depending on environmental factors, timing etc.).
Putting together 70's and 2017 data to draw inference on mortality does not seem valid; a comparison would be great though, but it not performed here.
There is not clear and satisfactory explanation of why snakes that are slightly larger (across a range of lengths) are more likely to be killed. I understand if the largest animals would be targeted by drivers, as I have witness firsthand myself (large, beautiful Timber Rattlesnake ran over in Pennsylvania by a pickup truck that went out of its way to do it; I was driving right behind the truck; people suck), but not why this would be the case across a range of sizes. I suspect that this is simply an artifact of imperfect detection of AOR and a small sample size. As such, placing too much emphasis on this analysis and using it to predict selection is very concerning.

Selection work: this is difficult to evaluate because of the weak statistical support based on imperfect data. As an ecologist, there seems to be a huge leap from a handful of snakes dear or alive o road to making claims that the snake population should shrink by 12 cm in average (also based on using a simple formula with coefficients that may or may not be relevant here). In Discussions, the author compares their findings to other studies of selection on morphology, but those studies are long term, have large sample sizes etc., which is not the case for this study.

Additional comments

Overall, I advise the authors to rethink their study, identify the main take home messages and the assumptions behind using many sources of data, shorten the Results and re-evaluate the selection component.
This could be a great snake road ecology study if the author (1) clearly explains how each dataset is relevant and used in various analyses, (2) re-write various sections of the paper to highlight that AOR cannot be taken at face level due to imperfect detection, thus the predictions are likely to be biased, (3) remove the selection study which is very much a stretch

Line 54: how low are these levels?
Line 80-83: This is a very strong and bold statement. I assume asserting non-random mortality based on such little data (and with likely a very wide margin of error) would be difficult. I am not an evolutionary biologist, but if selection against large body size exists, that would be manifested at the scale of the entire population. Thus, without a 'control' population not affected by roadkill, this would be difficult to prove, especially since selection has presumably been underway for a long time (since the road was constructed).
Line 130-131: Comparison across al pooled data? What about comparison between the old and new data? Wouldn't that provide a better evidence for selection over decades (if indeed this selection against large body size exists)
Line 137-138: Is this data from 70's and 2000's combined? That sounds like a stretch since an Interstate Highway will likely have much higher mortality, there have been changes in the population in terms of frequency of different sizes because of road mortality etc.... There are many issues with combining these data and I don't see any limitations mentioned here. Plus, the abstract only mentions 101 snake encounters
Line 138: This entire analysis is based on 48 snakes? How many DOR and how many AOR? Given the small sample size (and potentially uneven between AOR and DOR), I doubt that you can fit a model with more than 2-3 parameters. Adding interactions as mentioned seems untenable because you simply lack the statistical power. Other issues: is month considered as a continuous variable? It sounds like you should use a quadratic term since there is no expectation of a linear relationship. In any case, more detail is needed here to evaluate the correctness of this analysis and its statistical power.
RESULTS section: This section is very very long. The separate sections could be greatly reduced and only the most important patterns presented, with tables and figures to support them. Right not this entire section reads like an onerous report.
Line 190-192: I am not sure why this data is added here. It seems tangential to the study. perhaps useful to compare between the current data with 40 years ago and infer the potential for selection, but otherwise not that useful.
Section "Snake body sizes": Is this data combined across all years? Why is the author not comparing between 70s and now? This seems like the most relevant result, bot the difference between systematic and opportunistic. These results can be summarized in a couple of sentences with reference to figures or tables
Section "Size based mortality in c. atrox": Ok, some questions: what is the predictive power of this model? Did you calculate AUC or some other goodness of fit metric? The sample size are very small and the variation in lengths is great... just because these results were marginally signficant does not mean that the model has predictive power, especially since it is likely overparameterized.
Line 252-259: Interesting, but what is the relevance here? Mortality at population level is typically expressed as a function traffic volume, not 1/2 car scenarios. How does this fit in with the rest of the analysis
Line 262-271: I am not an evolutionary biologist, but I am having a difficult time reconciling these results with the data that this is based on (a handful of snakes in one single location that has been affected by this road for decades?). Besides issues of imperfect detection of snakes on roads (particularly the AOR snakes) and that a dead snake will be there to be found, while an alive snake would not, thus lending to an apples to oranges comparison, this analysis also implies that 40-50 years ago, the snakes were larger since this so-called selection has been acting on the population since it was first bisected by a road.
Thus logically, one could back-cast and calculate the 'real' length of snakes. Fg 6 is not very convincing since the range of these projections is huge.. In the absence of validation data from decades ago showing that snakes were larger back in the day, these projections are pure speculation based on very little data.
It is a huge leap to go from from 1 yr of systematically collected data on <50 snakes to predicting evolutionary change due to mortality pressure. I am sure the models can tell whatever, but validating it is a different story. The author presents some famous examples of selection in the discussion, but does not discuss the type and amount of data that led to those findings. In addition, how does the author believe that this shrinkage will happen morphologically and developmentally? Will snakes love vertebrae? Will they just have smaller vertebrae? For such a strong assertion, evidence that this can actually happen should be presented. Any other species undergoing such evolutionary processes? What makes this system unique? Given the data presented here, this should be prevalent globally, not just on a 30 mile stretch of road in NM.
The differences between AOR and DOR snakes could simply due to be randomness, particularly given the small sample size. Relying on a low-data model with a significant term at p=0.04, is a bit of unsettling for drawing conclusions about selection.
Line 368-372: issues with this interpretation: small DOR animals are harder to detect on roads because of the "tire" effect. Larger animals are more rare because older and larger animals typically make up a smaller portion of the populaiton (i.e., their survivorship decreases). As such, these interpretation seems to be biased by (1) a detection process and (2) a lack of understanding of population ecology and life tables

Reviewer 3 ·

Basic reporting

The manuscript adequately narrates the purpose of documenting and evaluating the impact of vehicles on snake mortality. It describes and compares species richness as well as the abundance and frequency of animals found alive and dead along a road transect in the northern Chihuahuan Desert. It identifies the most vulnerable species and suggests some possible causes, while also preliminarily modeling the potential evolutionary impact of vehicle collisions on the selection of smaller phenotypes in certain species. The manuscript appropriately considers methodological limitations and clearly and relevantly discusses the results and their potential biases. Overall, I consider it a well-managed, presented, and discussed research study.

Experimental design

The analysis of the data and the use of statistical methods also appear appropriate. Although the modeling analysis seems to show more biases because it considered only data from road-killed organisms, sampling off-road could have been conducted to assess the representativeness of sizes compared to the general sizes of snakes in the same population but outside the roads. However, this is a separate point in the research, and it is reasonable not to consider it given the way the study was conducted. In this sense, the discussion and interpretation offered regarding the results are well addressed and thoughtfully argued. I understand that the more space a snake occupies (due to its size) on the road, the more conspicuous it is and the higher the chances of being hit. This is a somewhat logical factor given the scenarios proposed by the author of the study. Nonetheless, the way of addressing the analysis is also appropriate and is substantiated with the analyses performed.

Validity of the findings

The analyses performed appear appropriate, are well justified, and are interpreted in a manner consistent with the generated evidence.
The results are well discussed. Overall, the document offers adequate reproducibility to continue the study in other locations and expand the comparative strategy.

Additional comments

The manuscript narrates the purpose of documenting and evaluating the impact of vehicles on snake mortality. It describes and compares species richness as well as the abundance and frequency of animals found alive and dead along a road transect in the northern Chihuahuan Desert. Its very informative and also offers a nice document to evaluare in a comparativa way with other placer arround the Chihuahuan Desert. I Think the manuscript is well developed to me published.

I Gad only one questions: Line 130. When you said Using all data… do you mean both studies? Yours and Price and LaPointe data?

Reviewer 4 ·

Basic reporting

No comment

Experimental design

This manuscript aims to demonstrate that vehicle collisions are a source of non-random mortality using several species sampled across a controlled time period on a stretch of road in New Mexico, USA. The study suggests that roads act as a source of directional selection against larger-bodied individuals, leading to a potential reduction in overall body size over time. While this is a compelling hypothesis, the evidence presented does not support the strength of the conclusions drawn. There are significant issues with the interpretation of the data, analytical framework, and structure of the manuscript that ultimately undermine the central thesis. I will provide an outline of my primary concerns with this manuscript below.

Validity of the findings

1. The primary claim, that larger snakes are more likely to be killed on roads, implying selection against body size, is potentially confounded by strong covariation between size and sex. In rattlesnakes and many other snake species, males are generally larger than females and exhibit greater movement, especially during the breeding season. Increased movement raises the probability of road encounters, making males disproportionately likely to be observed and killed on roads. Such pattern is evident in the data: male C. atrox outnumber females nearly 3:1 (n = 47 males, n = 17 females), and larger SVL and TL values are heavily skewed toward males. Although the manuscript notes that sex was not a statistically significant predictor of DOR (95% CI for odds ratio includes 1.0), this does not justify minimizing its biological relevance. Sex is potentially collinear with both body size and road use probability. Moreover, model comparisons (Table 4) show that interaction models including sex × total length are within ΔAIC < 1 of the top model, indicating that size effects may differ by sex. This biologically meaningful interaction is neither interpreted nor addressed in the discussion. As a result, the conclusion that selection is acting on size independently of sex is not supported. Robust inference about size-biased selection requires explicitly testing and interpreting interaction effects involving sex.
2. The manuscript goes beyond correlational analysis and simulates future evolutionary change, projecting a reduction in body size over multiple generations due to road mortality. However, these simulations rest on the untested assumption that observed mortality is driven by directional selection on size as heritable trait using metrics from another viper species. In this context, simulations are speculative and not empirically justified. While such projections may be useful as exploratory tools, they should not be framed as evidence of evolutionary response. As written, they contribute to an overextension of the manuscript’s conclusions.
3. Although the study includes multiple snake species, only C. atrox is analyzed in the context of mortality predictors and selection. Other species are included primarily descriptively, with no statistical analysis of DOR rates relative to body size. This raises concerns about scope and focus: if the manuscript’s aim is to generalize about road-induced selection, then comparative analyses should be conducted on additional species with sufficient sample sizes, particularly if the additional species are included elsewhere in the study. If that is not feasible due to sample size, the manuscript should be narrowed in scope to focus explicitly on C. atrox. As it stands, the inclusion of other species appears to broaden the dataset without contributing to the core analytical or evolutionary questions.
4. Male-biased sampling is evident not only in C. atrox but across several of the other species in the dataset. This likely reflects well-documented sex-biased movement patterns in snakes, where males travel farther and more frequently during reproductive periods (Reinert & Zappalorti, 1988; King & Duval, 1990). These behaviors increase exposure to roads and inflate male representation in roadkill data. Without accounting for this ecological bias, any relationship between size and DOR is confounded by male overrepresentation. The manuscript does not critically examine this issue, even though it directly impacts the interpretation of size-related mortality and the claim of directional selection.

Citations:
Reinert, Howard K., and Robert T. Zappalorti. "Timber rattlesnakes (Crotalus horridus) of the Pine Barrens: their movement patterns and habitat preference." Copeia (1988): 964-978.

King, Michael B., and David Duvall. "Prairie rattlesnake seasonal migrations: episodes of movement, vernal foraging and sex differences." Animal Behaviour 39.5 (1990): 924-935

Additional comments

The hypothesis that roads drive morphological evolution in snake populations is worth exploring, but this manuscript does not provide sufficient evidence to support that claim. The central conclusion, that larger body size is being selected against due to road mortality, is undermined by unaddressed confounds with sex and speculative evolutionary projections. Furthermore, the inclusion of other species is not analytically justified and does not add to the study's core focus.
For these reasons, I do not recommend this manuscript for publication in its current form. The author may be able to develop a more appropriately scoped manuscript with a focus on descriptive road ecology or species-specific patterns, but the evolutionary claims should be removed or substantially revised with appropriate controls and caveats. My recommendations to the author are:
Explicitly test and interpret sex × size interactions in the statistical models. While sex was included as a factor in many models, it was treated primarily as a control variable rather than a central biological component of the pattern. Given the strong biological link between sex, movement behavior, and body size, it is important that interactions between sex and size be fully explored and reported. If males are driving both the size distribution and road mortality patterns, the claim of selection on body size becomes untenable.

Clarify the role of sex in the interpretation, not just the modeling. Although sex appears in the best-supported models, the discussion does not sufficiently consider sex-biased behavior as an explanation for size-associated mortality.

Either incorporate multi-species comparisons or remove them from the analysis. The current inclusion of multiple snake species mainly serves only a descriptive role, but the central analysis and conclusions are focused exclusively on C. atrox. If the author intends to make broader claims about the evolutionary consequences of roads, they should include parallel analyses on other species with sufficient sample sizes from their dataset.

---

## Round 0.2 · Minor Revisions

I ask you to listen to the reviewer's comments. You ignored many of them, making only minor changes to the manuscript. I hope that the new version of this article will be a radical improvement.

Reviewer 1 ·

Basic reporting

no comment

Experimental design

no comment

Validity of the findings

no comment

Additional comments

no comment

Reviewer 2 ·

Basic reporting

no comment

Experimental design

no comment

Validity of the findings

I am still skeptical of the inference on evolutionary pressure on body change based on these data; I am not an evolutionary biologist though, so I will let the other reviewers evaluate this aspect. The author mentions that drastic changes have been made to the Discussions about moderating the stements about projected evolutionary change, but they do not seem that major, except for a few sentences on limitations. Acknowledgment of the limitations does not equate to improving inference.

Additional comments

Thank you for clarifying many aspects of the methodology and data used for various analyses and implemeting test to evaluate model fit. An AUC of 0.75 is not that great, so I would be less assertive about the findings.
I am still skeptical of the inference on evolutionary pressure on body change based on these data; I am not an evolutionary biologist though, so I will let the other reviewers evaluate this aspect.
The author mentions that drastic changes have been made to the Discussions about moderating the stements about projected evolutionary change, but they do not seem that major, except for a few sentences on limitations. Acknowledgment of limitations does not equate to a improving inference. As such, retaining the title and general direction of the evolutionary biology aspect of the work is still misleading (also highlighted by another reviewer). I understand that this is a data-limited study, like many other snake studies. It is important data to present and publish, but one still needs to be cautious about extrapolating from such a small sample size over a small sampling window.

---

## Round 0.3 · accepted · Accept

Dear Dr. Hughes, I am pleased to inform you that this article has been accepted for publication.

Reviewer 2 ·

Basic reporting

no comment

Experimental design

no comment

Validity of the findings

no comment